# Gold Nanoparticles as a Biosensor for Cancer Biomarker Determination

**DOI:** 10.3390/molecules28010364

**Published:** 2023-01-02

**Authors:** Chien-Hsiu Li, Ming-Hsien Chan, Yu-Chan Chang, Michael Hsiao

**Affiliations:** 1Genomics Research Center, Academia Sinica, Taipei 115, Taiwan; 2Department of Biomedical Imaging and Radiological Sciences, National Yang Ming Chiao Tung University, Taipei 112, Taiwan; 3Department and Graduate Institute of Veterinary Medicine, School of Veterinary Medicine, National Taiwan University, Taipei 106, Taiwan

**Keywords:** gold nanoparticles, biosensing, surface plasmon resonance, cancer marker, surface modification

## Abstract

Molecular biology applications based on gold nanotechnology have revolutionary impacts, especially in diagnosing and treating molecular and cellular levels. The combination of plasmonic resonance, biochemistry, and optoelectronic engineering has increased the detection of molecules and the possibility of atoms. These advantages have brought medical research to the cellular level for application potential. Many research groups are working towards this. The superior analytical properties of gold nanoparticles can not only be used as an effective drug screening instrument for gene sequencing in new drug development but also as an essential tool for detecting physiological functions, such as blood glucose, antigen-antibody analysis, etc. The review introduces the principles of biomedical sensing systems, the principles of nanomaterial analysis applied to biomedicine at home and abroad, and the chemical surface modification of various gold nanoparticles.

## 1. Introduction

The unique optical properties of gold nanomaterials were revealed as early as the Middle Ages, when the scientist Michael Faraday prepared a gold colloid solution via the wet chemical synthesis method and demonstrated its extraordinary optical properties in 1857 [1], suggesting that gold particles (AuNPs) appear red in the nanoscale and gradually change to dark blue as the particle size increases.

It has been demonstrated that the incident light’s electromagnetic field can excite the surface-free electrons of metallic materials on the nanoscale, as the particle size is much smaller than the incident wavelength. The frequency of the incident light produces a collective oscillation motion called surface plasmon resonance (SPR), as shown in Figure 1a [2]. The uneven distribution of electrons illustrates the electric field at different cross-sections when free electrons are affected by electromagnetic radiation. For example, deviation from an electric field causes negatively charged electrons to move in the direction of the electric field. Transiently induced dipoles are generated, resulting in the separation of free electrons from the metal nucleus. This separation leads to a Coulomb restoring force in the opposite direction and, finally, causes the metal’s free electrons to generate a collective back-and-forth oscillatory motion at both ends of the particle. Surface plasmon absorption bands appear in the ultraviolet and visible wavelengths of light as a result of this resonance phenomenon. This explains the different colors of metallic nanomaterials under white light. As shown in Equation (1), there is a strong correlation between the peak of surface plasmon resonance (ω_sp_) and the surface charge density of the material. In contrast, the charge density of the particle is affected by particle size, shape, structure, dielectric constant, etc. [3].

In addition to the above advantages, AuNPs combined with composite materials of upconversion nanoparticles can solve some of the shortcomings and increase the diagnostic and therapeutic value of single materials. For instance, AuNPs need light energy to promote SPR, which upconversion nanoparticles can provide. As shown in Figure 1b, Sun et al. [4] used a AuNP dimer as a core material. They combined it with a grafted photosensitizer to form a core-satellite type composite material for diagnostic and mouse tumor treatment. A 980 nm laser was used to excite the upconverted nanoparticles to emit fluorescence to drive the photosensitizer for photodynamic therapy. An 808 nm laser was then used to excite the AuNP dimer for photothermal treatment. Using near-infrared light as the excitation source can increase the penetration of biological tissues and reduce unnecessary heat damage. A combination of photothermal and photodynamic therapy is more effective than a single light treatment. In addition to its therapeutic function, the above-converted emitted light can be used as a fluorescent probe for cellular calibration. In nuclear magnetic resonance imaging, gadolinium (Gd^3+^) can serve as a contrast agent (NMRI) to enhance the bio-imaging function. Due to the photothermal conversion effect, the excited AuNPs can be used for photoacoustic imaging (PAI), which provides an alternative way to image tumors. The composite material can also be used as a contrast agent for computed tomography (CT) imaging to enhance image contrast. Therefore, the composite material has excellent therapeutic effects and high application value for four types of bioimaging calibrations. Based on the advantages mentioned above, AuNPs are often used in biomedical research to carry certain substances into organisms. The surface of AuNPs can be coated to modify the specific molecules and sensors expected to be held on their surface when the AuNPs are used carry biomarker receptors into a body. Optical biomarkers are produced by SPR using AuNPs adhered to high-refractive-index surfaces. The AuNPs absorb laser light and generate electron waves on their surface to generate measurable signals for any analyte bound to the AuNPs. Therefore, this review is focused on AuNPs, specifically considering the changes in signals caused by the characteristics of AuNPs that can be used to detect molecules, such as DNA, RNA, and protein, flowing into an organism. This review further discusses the synthesis and properties of AuNPs, as well as their potential applications as sensors.

## 2. Synthesis of Gold Nanoparticles

Depending on the method of preparation, AuNPs can be classified into two major categories: (i) top–down and (ii) bottom–up. The top–down approach includes the template method [5], lithographic methods [6], and catalytic methods [7]; the bottom–up process consists of the electrochemical method [8], the seedless growth method [9], and the seed-mediated growth method [10]. It is most commonly used to synthesize AuNPs by seed-mediated growth.

### 2.1. Surfactant-Preferential-Binding-Directed Growth

The seed-mediated growth method was first proposed by Wiesner and Wokaun [11] in 1989 to obtain the crystals of AuNPs by reducing the tetrachloroauric acid (HAuCl_4_) with phosphorus elements using the crystals as the nuclei to grow AuNPs. In the beginning, Turjevich’s team used sodium citrate as a reducing agent and an interfacial reactive agent to reduce HAuCl_4_ from Au^3+^ to Au by hydrothermal method to synthesize many gold nanoparticles of different sizes under different reaction parameters. In the aqueous phase, the AuCl_4_^−^ atoms are reduced by sodium citrate to form gold atoms, which then aggregate to form gold nanoparticles. The negatively charged citrate ions play the reducing agent and capping agent. The size of the particles is controlled by the ratio of gold ions to sodium citrate with the heating time of the reaction. At this time, the gold nanoparticles are protected by the negatively charged citrate on the outside of the particles and are stably stored in the aqueous solution. In 2003, Nikoobakht and El-Sayed [12] proposed two modifications of this synthesis method: sodium citrate, the protective agent, was replaced by hexadecyltrimethylammonium bromide (CTAB) in the synthesis of the seeds and Ag^+^ was used to regulate the aspect ratio of the AuNPs. After modification, the yield of AuNPs produced by the crystal growth method could reach 99%, and the aspect ratio of AuNPs could be adjusted from 1.5 to 4.5 by changing the silver ion concentration. There are two significant steps in the crystal growth process. The first step is the preparation of a crystal solution; a tetrachloroauric acid solution containing the surfactant CTAB is reduced to crystals with the reductant sodium borohydride (NaBH_4_), in which the surfactant CTAB is used as a protective agent to stabilize the crystal. In the second step, reductant ascorbic acid is added to a growth solution containing the surfactant CTAB, silver nitrate (AgNO_3_), and tetrachloroauric acid, and the trivalent gold ions of tetrachloroauric acid in the solution are reduced to monovalent gold ions. Then, the crystalline solution synthesized in the previous step is added to the growth solution to produce AuNPs. Currently, Murphy et al. propose a mechanism in which surfactant-preferential-binding-directed growth is responsible for the growth of AuNPs [13,14]. This team observed the structure of AuNPs and the effects of various reaction conditions on AuNP generation with high-resolution penetrating electron microscopy. The growth of AuNPs begins with the change of the intrinsic structure of the crystalline species, and the disruption of the crystalline species symmetry leads to non-isotropic growth, in which the {100} lattice of the AuNPs has a higher surface energy. Hence, CTAB preferentially adsorbs on the {100} surface. In addition, the adsorption of CTAB stabilizes the {100} surface and restricts the growth of the {100} surface so that the AuNPs grow towards the ends of the {111} surface to form a rod, as shown in Figure 2a.

### 2.2. Electric-Field-Directed Growth Mechanism

According to Mulvaney et al., the electric-field-directed is the second growth mechanism [15]. This mechanism involves the addition of AuCl_4_^2−^ to the CTAB microbattery, followed by its reduction to an AuCl_2_^−^-CTAB microbattery by a reducing agent, and the reaction equation is as follows. The negatively charged AuCl_2_^−^ at the surface of the CTAB microcell attracts the CTAB on the surface of the crystal, causing the AuCl_2_^−^-CTAB microcell to collide with the CTAB-protected crystal. In contrast, the electrons on the surface of the crystal are transferred to the monovalent gold ions to produce the reduction reaction. Finally, the collision probability of the two ends of the crystal becomes greater than that of the side, resulting in the generation of the rod-like structure. In the synthesis of AuNPs, the silver ions of silver nitrate are used to regulate the aspect ratio of AuNPs by forming silver bromide (AgBr) with the bromine ions (Br-) of CTAB to reduce the charge density and the negative electric repulsion between the functional groups at the head end of the CTAB; therefore, regulating the concentration of silver ions can regulate the arrangement of the soft templates of CTAB and change the aspect ratio of AuNPs. As shown in Figure 2b, As part of the growth mechanisms of AuNPs, CTAB plays a crucial role in preventing AuNPs from aggregating due to electrostatic repulsion between positive surface charges [16].

## 3. Biological Characteristics of AuNPs

Metal materials have localized surface plasmon resonance (LSPR) properties, and gold nanomaterials have become a popular research material because of their unique optical properties. To understand the LSPR phenomenon, the SPR effect first needed to be understood. In 1902, Wood [17] designed a grating experiment on polarizing light. He found an abnormal phenomenon of diffraction on the flat metal surface when the grating was placed near a metal surface. Later, Fano provided a theory to explain this, i.e., light is a kind of electromagnetic wave that could interact with the free conduction electrons on the surface of the metal. In this case, plasma polaritons propagate along the metal-dielectric interface in both directions. This is called surface plasma polarization (SPP) or surface plasma resonance (SPR), as shown in Figure 1c [18]. Furthermore, LSPR occurs at the nanoscale because the incident light is far greater than the nanomaterials, resulting in all the free electrons on the surface reacting together. Briefly, when light is irradiated on a metallic nanoparticle, the free conductive electrons on the surface interact with the incident light via the electromagnetic interaction, followed by all polarized electrons oscillating together at a specific frequency, as shown in Figure 1d. In 1908, this phenomenon in spherical nanomaterials was investigated and resolved by Mie using Maxwell’s equations to explain the relationship of the extinction (SPR) spectra (extinction = scattering + absorption) in any particle size [19].

### 3.1. Localized Surface Plasmon Resonance (LSPR)

As a result of some elemental species inside organisms, UV and visible light have limited penetration abilities regarding skin, such as water and oxy/deoxyhemoglobin, which show tremendous individual absorption ranges from ~900 to 1000 nm and from ~400 to 690 nm, respectively [20]. Accordingly, the suitable SPR peaks at the greatest wavelength are between 700 and 900 nm, which is intended to mean that the energy is more efficiently absorbed when moving away from the red-shifted region to the near-infrared (NIR) region. Gold nanorods with an anisotropic structure generate two bands of SPR at 520 nm or 650~1200 nm, denoted as the transverse and longitudinal axis, respectively. It is easy to red-shift the long axis of an SPR band by increasing the aspect ratio for gold nanorods. When AuNPs are excited by incident light to generate surface plasmon resonance, the oscillation directions of surface free electrons are divided into two types due to different polarization modes, one of which is the oscillation motion along the long axis of the material, and the other is the oscillation motion along the short axis of the material [21,22]. The short-axis surface plasmon oscillates as a result of the short-axis surface plasmon, and AuNPs exhibit a characteristic absorption peak at about 520 nm in the visible light range. This is called the short-axis absorption peak (transverse band). The long-axis plasmon oscillations produce stronger absorptions and fall in the longitudinal band, which corresponds to the long-wavelength region. The short-axis absorption peaks do not significantly change with the size change of AuNPs. On the contrary, the long-axis absorption peaks drastically change with the aspect ratio (length/width, aspect ratio) of AuNPs. When the aspect ratio increases, the long-axis absorption peak shifts from the visible light band to the near-infrared light band, resulting in a red-shift phenomenon.
(1)$$\omega_{\mathrm{sp}}=\sqrt{\frac{{\mathrm{ne}}^{2}}{m_{0}\varepsilon_{0}\left(\varepsilon_{\infty }+{\;\kappa \varepsilon }_{m}\right)}-\Gamma^{2}}$$

ω_sp_: surface plasmon resonance wavelength; n: free carrier density; m_0_: effective mass of charge carrier; ε_0_: dielectic constant offree space; ε_m_: medium dielectric constant; ε_∞_: high dielectric constant of medium; κ: geometry of nanomaterials; and Γ: damping constant.

When considering using AuNPs for biotherapeutic applications, it is essential to use appropriate-length AuNPs because red blood cells and water occupy a large proportion of the biological body. In the visible wavelength (<650 nm) and infrared region (>900 nm), the absorption intensity of red blood cells and water molecules is the highest. Still, the absorption coefficients of red blood cells and water are the lowest in the near-infrared wavelength between 650 nm and 900 nm. This distance is also called the near-infrared (NIR) window [20]. Therefore, when near-infrared light is used as the excitation source, the biological tissues absorb the least, which also means that the penetration depth of near-infrared light in biological tissues is higher than that of other wavelengths and is most suitable for biomedical applications. When the aspect ratio is about 4, the position of the long-axis absorption peak of AuNPs is red-shifted to about 800 nm, which is the near-infrared region (650–900 nm) where the incident light penetrates deepest into human tissues. AuNPs can convert near-infrared light energy into heat when stimulated by near-infrared light [23,24,25]. As a result of the scattering and absorption of light, the scattered light is converted into thermal energy when the incident light absorbs it, AuNPs can be used for the detection of diseased tissues and the diagnosis and treatment of tumor cells at the same time; therefore, AuNPs have great potential in the biomedical field due to their unique optical properties.

### 3.2. Surface Modification of Gold Nanoparticles

AuNPs are often used as emerging nanomaterials with unique and excellent optical properties. They are widely used in the biotechnology and biomedicine fields, including biomedical imaging, drug transportation, disease diagnosis, and treatment, as shown in Figure 3a. Whether AuNPs are synthesized by the crystal-free growth or crystal growth methods, their surface must be adsorbed with a CTAB double-layer protective agent so that the AuNPs are monodisperse in aqueous solutions due to electrostatic repulsive forces. In addition, the CTAB can protect AuNPs in environments with high concentrations of salts. However, CTAB causes a high degree of biotoxicity when it is detached from the surface of AuNPs. Therefore, the surface modification will become an essential issue if we want to apply AuNPs in the biomedical field further. Currently, there are three types of surface modification methods for AuNPs.

#### 3.2.1. Ligand Exchange

The differences between carboxylic acids and amines can be observed when compared to other functional groups and alcohols, and there is a strong covalent bonding between thiol and gold in AuNPs. The molecules to be modified on the surface of AuNPs can be functionalized with thiol functional groups, and the thiol groups can generate Au-S bonds on the surface of the AuNPs to replace the interfacial reactive agent CTAB, deoxyribonucleic acid (DNA) [26], or other small molecules [27]. For example, Maltzahn et al. [28] replaced CTAB with PEG of high biocompatibility on the surface of AuNPs to enhance their stability, improve the shortcomings of CTAB-coated AuNPs that tend to aggregate in serum environments, and increase the circulation time (t1/2) of the material in an organism for up to 17 h, as shown in Figure 3b. Due to the close arrangement of CTAB on AuNP surfaces, some biomolecules, including antibodies and proteins, cannot bind to them after they have been modified with thiols. Therefore, small molecules, such as 3-mercaptopropionic acid (MPA) and 11-mercaptoundecanoic acid (11-MPA), must be used first. Then, the functionalized ends of these molecules can be used to covalently bond with the large molecules, such as antibodies or proteins, to be modified. For example, Yu et al. [29] first replaced CTAB on the surface of AuNPs with MUDA and AMTAZ, two small molecules of organic thiol, to significantly reduce the biotoxicity of the composite, which then reacted with the amine or carboxylic acid groups of MUDA and AMTAZ to form an amide bond with the antibodies. Antibody-modified AuNPs can be applied to nanosensors and target therapy.

#### 3.2.2. Electrostatic Adsorption

Furthermore, gold surfaces can be modified by forming gold-sulfur bonds between thiol groups and gold surfaces, as well as electrostatic adsorption between negative molecules and positive CTAB, while antibodies or proteins may also be directly adsorbable onto the surface of AuNPs [30] or via layer-by-layer modification [31]. The layer-by-layer modification of polymers on the surface of AuNPs can easily manipulate their surface properties. For example, Gole et al. [32] performed the layer-by-layer modification of negatively and positively charged sodium polystyrene sulfonate (PSS) and poly (dimethyl diallyldimethylammonium chloride) (PDADMAC), respectively. The polymers on the surface of the AuNPs were selectively attached to the substrates with different charges, as shown in Figure 3c. In addition, the modification of polymers with further charges also affects the phagocytosis behavior of cells [33]. Compared with the gold–sulfur bond modification strategy, electrostatic adsorption does not require ligand substitution and is relatively simple, fast, and efficient.

#### 3.2.3. Electrostatic Adsorption

In addition to using the two abovementioned methods to improve the stability and applicability of AuNPs, Sendroiu et al. [34] proposed covering the surface of AuNPs with a layer of silica to strengthen their application value in living organisms through silica’s excellent biocompatibility and ease of modification, which also enables the use of application strategies. In addition, it is possible to increase the surface area of AuNPs by using mesoporous silica as a carrier, which can facilitate the loading of more drug molecules and provide the material with more functions. For example, Zhang et al. [35] used mesoporous silica-coated AuNPs to load the anticancer drug doxorubicin (Dox) and modulate the long-axis absorption peak of the AuNPs to the near-infrared region so that the AuNPs could undergo photothermal conversion after near-infrared laser excitation to produce hyperthermia while also promoting the release of Dox loaded into the mesopores to enhance chemotherapy (chemotherapy). The two-photon fluorescence generated by the Dox release and the AuNPs could be used for cell imaging. This nanoplatform could simultaneously realize the dual effect of cell imaging and therapy, as shown in Figure 3d.

### 3.3. Biosensing Assays

#### 3.3.1. Förster Resonance Energy Transfer (FRET)

An energy transfer mechanism based on Förster resonance energy transfer (FRET) involves energy transfer between two chromophores. FRET is similar to near-field transport, i.e., compared with the wavelength of the excitation light, the reaction distance of action is much smaller. Near-field regions are characterized by the emission of virtual photons by excitations of donor chromophores, which are then absorbed by acceptor chromophores. Due to the fact that these photons violate energy conservation and momentum, these particles are not detectable. Therefore, FRET is considered a radiation-free process. From quantum electrodynamics calculations, we can determine the short-range and long-range approximations of radiation-free and radiative energy transfers under a unified field [36,37], respectively. When both chromophores are fluorophores, the mechanism is called fluorescence resonance energy transfer; however, in reality, the energy transfer is not conducted through fluorescence [38,39]. Since this phenomenon is based on non-radiative energy transfer, we prefer to use FRET terminology to avoid misleading names. It should also be noted that FRET is not limited to fluorescence as it can also be related to phosphorescence. For instance, a combination of AuNPs and upconversion nanoparticles was first applied to enhance the fluorescence intensity of upconversion nanoparticles with the unique SPR optical properties of AuNPs. After continuous research, in 2015, Zhan [40] and his research team adjusted the aspect ratio of AuNPs so that the absorption peaks of the long and short axes overlapped with the excitation source and emission position of the upconversion nanoparticles, and controlled the particle size of the upconversion nanoparticles to 4 nm to ensure that all the doped ions were affected by the electromagnetic field of the SPR to enhance the intensity of the upconversion emission. Dadmehr [41] et al. conjugated the aptamers to deposit on the surface of graphene oxide decorated with AuNPs, and the following formation of a hetero-duplex stem-loop structure led to fluorescence quenching. Moreover, Hu et al. [42] also reported a sensitive fluorescent probe with off–on FRET response for harmful thiourea that can be detected by fluorescent carbon nanodots combined with AuNPs via electrostatic interaction. As shown in Figure 4a, the electromagnetic field generated by the long-axis absorption peak of AuNPs very strongly absorbs photons, which consequently enhances the absorption rate of photons via the surrounding upconversion nanoparticles, thus increasing the intensity of the emitted light. The short-axis absorption peak of AuNPs enhances the density of the optical states of upconversion nanoparticles through the Purcell effect, which regulates the radiative decay rate to strengthen the intensity of the upconversion.

#### 3.3.2. LSPR Electric Field Enhancement

AuNPs enhance the light emission of organic fluorophores by means of the SPR effect. In this article, we discuss this effect using an example of plasmon-enhanced fluorescence emission [43]. As a result of the abundance of conductive electrons in AuNPs, both the intensity and the rate at which electromagnetic fields are radiated increase as an electromagnetic field is excited. However, AuNPs could simultaneously quench or enhance the emitting intensity because it also has a high absorption coefficient. Therefore, the distance between AuNPs and another emitter, which we call upconversion nanoparticles (UCNPs), is the key point to control which behavior in the significant part. In Figure 4b, there is another viewpoint that an emitter with a low quantum yield could be significantly enhanced compared with an emitter with a higher quantum yield, even following radiative rate enhancement, meaning that UCNPs have a high potential for enhancing upconverted luminescence efficiency based on the low quantum yield of UCNPs. Among the plasmonic nanomaterials, silver and gold have been studied extensively for their ability to enhance upconversion. Silver has a more substantial SPR effect than gold because overlapping interband transitions do not dampen its plasmon resonance. In spite of this, silver does not consistently result in a greater enhancement of upconversion emissions, suggesting that other factors such as overlap between spectral bands or overlap between fields may have a greater influence on the observed enhancements. Additionally, the maximum enhancement for a core-shell or flat gold film structure is 10 times lower than that of a gold hole array-patterned system with 450 times enhancement because SPR displays a specific region with a higher resonance strength for two reasons: the coupling phenomena between each AuNP and SPR’s focus on sharp tip structures.

Additionally, AuNPs be used as energy absorbers because of the significantly different order of extinction coefficient between AuNPs and organic fluorophores. The value of the extinction coefficient for the commonly used organic dye indocyanine green (ICG) is only 10^5^ (cm^−1^/M) at 790 nm, but AuNPs have an extinction coefficient of 10^11^~10^13^ (cm^−1^/M), which is higher than the ICG 6~8 order. Therefore, AuNPs can be used to improve ROS production from ICG by accepting the energy transferred from the dye [44].

#### 3.3.3. Fluorescent and Colorimetric Assay

AuNPs have unique physical and chemical properties, which make them easier to be novel chemical and biological sensors. Over the past ten years of research, AuNPs have been widely used as sensors and in detecting heavy metals. Among them, the colorimetric method is a common and convenient detection method. The content of the component to be tested is determined through the solution’s color depth, and the sample’s concentration is analyzed by optical and photoelectric colorimetry. Both methods are calculated through the Beer–Lambert law (Equation (2)). For example, DNA-functionalized gold nanoparticles can be used to detect lead ions with a colorimetric method: DNase-based functionalized gold nanoparticles; the sensor’s detection range can be from 3 nM to 1 μM [45]. However, DNA molecule synthesis and chemical modification are very complex and expensive.
(2)A=ε·l·c

A is the absorbance; ε is the molar attenuation coefficient or absorptivity of the attenuating species; *l* is the optical path length in cm; *c* is the concentration of the attenuating species.

However, colorimetric techniques tend to be less sensitive and less selective. To effectively improve its sensitivity and resolution, recent studies have used composite materials to improve the accuracy of colorimetry. In current literature, AuNPs are used as a heavy metal sensor at different pH values. Mercury ions and lead ions will form mercury-gold and lead-gold alloys on the surface of AuNPs [46]. In the presence \ of hydrogen peroxide (H_2_O_2_), the fluorescent substrate is catalyzed with peroxidase-like properties. In order to make the sensor more stable to heat, pH, and salt, Li [47] et al. designed a fluorescent and colorimetric dual-modal sensor based on AuNPs prepared using carbon dots (CDs) to detect the concentration of Cu^2+^ and Hg^2+^ in water. In addition to environmental heavy metal detection, AuNPs can also be used to analyze cancer biomarkers through dual-mode fluorescence colorimetry. Dadmehr [48] et al. used AuNPs grafted with gelatin and combined with gold nanoclusters to detect matrix metalloproteinase. Food samples can also be detected by AuNPs colorimetric method. Shahi [49] et al. used gelatin-functionalized AuNPs (AuNPs@gelatin) and applied a competitive colorimetric assay to detect aflatoxin B1 (AFB1) and measure the concentration of this carcinogen in food samples.

## 4. Clinical Biomarkers for Early Cancer Detection

Crick proposed the central dogma in 1957 and demonstrated that the transmission of genetic information relies on DNA, RNA, and proteins [50]. With the development of molecular biology technologies, people can now identify how diseases are caused by specific molecules involved in transcription and translation. It is becoming increasingly common to use these molecules as diagnostic markers for diseases related to cancer and other diseases. Nevertheless, the development of imaging and molecular marker-related technologies is considered the most effective method of improving the resolution and sensitivity of detection. Since AuNPs enable image transduction and are non-invasive, many studies have been conducted to label many key disease-related molecules identified via AuNPs, thereby improving the sensitivity of diagnosis (Figure 5).

### 4.1. DNA

DNA is one of the most critical substances cells used to store genetic information. Most commonly, it is found in the nucleus as a chromosome. Chromatin stability can be regulated by histone modifications, DNA methylation and DNA-histone interactions, the results of which is only gene expression, but not the intrinsic sequence of genes, known as epigenetic regulation [51]. Most eukaryotic cells, including mammals, regulate DNA methylation in GC-rich regions or CpG islands, primarily through DNA methyltransferases. Such modification leads to dysregulation and abnormal expression of specific genes. These expression imbalances have been linked to the development of several diseases [52]. It was therefore necessary to develop these specific variants of methyltransferases as a means of detection.

The detection of DNA methylation has been carried out using several techniques [53], including genome-wide methylation extent analysis, gene-specific methylation analysis, and screening for new methylated sites. Assays are restricted by restriction sites [54], including enzymatic hydrolysis-based traditional methods, such as high-performance liquid chromatography (HPLC), high-performance capillary electrophoresis (HPCE), restriction landmark genome scanning (RLGS), and methylation-sensitive arbitrary primed polymerase chain reaction (MS-AP-PCR), as well as restriction enzyme-free methods, such as methylation-specific PCR (MSP) and real-time methylation-specific PCR. However, many technologies may generate false-positive results and lack the sensitivity to detect them. Nevertheless, there are methods that can be used to improve this insufficiency [54], such as combined bisulfite restriction analysis (COBRA), a methylation-sensitive single nucleotide primer extension (Ms-SNuPE) assay, Ms-SNuPE assays, matrix-assisted laser desorption/ionization mass spectrometry (MALDI-MS), double-labeled probes, and quantitative PCR (qPCR), but these methods are not cost-effective.

The introduction of AuNPs is considered to be an effective method to improve these shortcomings. Upon exposure to methyltransferases, AuNPs modified by a double-strand DNA probe will change from a well-dispersed state to an aggregated state. By measuring the A620/A520 difference, a colorimetric assay based on AuNPs can be developed to detect the presence of methyltransferases in the environment [55]. It is also possible to apply the same logic by using surface-enhanced Raman spectroscopy (SERS) to detect single-nucleotide polymorphisms. Detecting changes in SERS was found to be enhanced by comparing the conversion of cytosine and 5-methylcytosine into sodium bisulfite with the binding difference between AuNP-dGTP-cy5-probe and sodium bisulfite [55]. According to Liu’s research, a polyadenine-DNA hairpin probe that is methylene blue-labeled and can be methylated by methyltransferases can be used as a restriction enzyme recognition tool for determining the differences in electrochemical response and DNA methylation [56]. Additionally, a specific probe designed to detect methylation combined with a fiber optic nanogold-linked sorbent assay can directly detect the methylation state of the tumor suppressor SOCS-1 through transmitted light intensity [57].

Furthermore, biomarkers that identify somatic mutations are also used to diagnose potential genetic diseases as a preventative measure. For example, the BRCA1 mutation is one of breast and ovarian cancer most commonly occurring oncogenes. It was found that a DNA capture probe immobilized in AuNP can be used to detect these discovered prognosis biomarkers, with a detection accuracy reaching femtomolar levels [58]. Additionally, an electrochemical DNA sensor based on a BRCA1 DNA sequence-based tetrahedral-structured probe was fabricated with AuNPs to precisely detect BRCA1 [59]. It has also been found that using Bi_2_Se_3_-AuNPs as a load signal probe to form a sandwich with BRCA1-immobilized silicon improves diagnostic accuracy [60]. It is also possible to use boron nitride quantum dot AuNPs to enhance the detection of BRCA1/2 [61]. An electrochemical method of prostate cancer-specific DNA sequences (PCA3) using chondroitin sulfate-AuNPs has been demonstrated to treat prostate cancer [62].

The cell-systematic evolution of ligands by exponential enrichment (cell-SELEX) is the process of discovering small oligonucleotides through arithmetic. It has been found that these less than 100-mer aptamers bind with a specific affinity to the cell surface through a unique folding structure and have applications in the detection and diagnosis of diseases [63], as well as in the annotation of metals [64,65,66]. The identification of cancer cells was found to be effective when combining specific aptamers with AuNPs and simultaneously forming complexes with magnetic beads and magnetic forces [67]. Using this method, it is possible to separate a sample using complementary sequence DNA (capture DNA) and determine whether the sample contains cancer cells via electrochemiluminescence. Furthermore, AuNPs have a good solubility, high surface reactivity, and excellent bioactivity. Taking advantage of their high fluorescence efficiency and versatility in surface modification, concatemer quantum dots were further used as a component of nanocomposites with AuNPs. This study demonstrated that MWCNTs@PDA@AuNP nanocomposites can improve aptamer recognition specificity [68].

Similar oncogenes such as KRAS can also be used as biomarkers for cancer diagnosis. MWCNTs–PA6–PTH conjugates can recognize mismatched bases in KRAS single-strand DNA-AuNPs with over 50% efficiency when used with KRAS single-strand DNA-AuNPs [69]. This method can also be applied to gastric tumors using KRAS as the target. A hairpin-DNA that recognizes KRAS mutations and conjugates with Cy3-AuNPs can be highly effective in identifying gastric tumors. As a result, KRAS-specific hairpin-DNA can go against the sequence of the KRAS sequence to inhibit its biological functions, which include vascularization and metastasis, resulting in an improvement in the survival rate [70].

Apart from forming nanocomposite materials with different elements, aptamer-AuNPs can also be used to separate circulating tumor cells from blood using microfluidics, with a 39-fold increase in efficiency [71]. According to this method, when using aptamer-AuNP probes for detection, one can determine the presence or absence of binding based on the color difference between the two probes [72]. It is therefore possible to apply similar assays to identify different molecules in liquid biopsies, such as circulating tumor DNA (ctDNA) and PIK3CA. In conjunction with peptide nucleic acid-AuNPs, the anti-5-methylcytosine antibody conjugate apoferritin can be used to specifically identify ctDNA in patient blood by detecting differences in ions with an accuracy of 50–10,000 fM [73]. The use of aptamers is helpful for not only identifying specific cells as a diagnostic tool but also targeting therapies. A Pt@Au nanoring@DNA nanocomposite was found to be helpful as a diagnostic and therapeutic strategy due to its photothermal therapy abilities for tumor cells when exposed to near-infrared light [74]. It has also been shown that combining tetrahedral DNA frameworks and AuNPs can further enhance the capture efficiency of BRCA1 ctDNA from 1 aM to 1 pM [75].

In addition to the DNA of a tumor itself, different genotypes of tumor-associated viruses have been designed as probes in conjunction with polyaniline-AuNP to develop biosensors that can identify the specific DNA targets of human papillomavirus, as well as other viruses, to improve diagnosis efficiency [76]. Studies have also shown trichomoniasis infections cause several cancers. The abovementioned probes can also be used detect trichomoniasis by designing a sequence that targets trichomoniasis and forming a biosensor with AuNPs [77].

### 4.2. RNA/miRNA

RNA messengers, also known as messenger RNA (mRNA), are essential molecules that are transcripted from DNA, transmit information, and serve as templates for translation into proteins. The use of dysregulated mRNA is also considered to be one of the most important cancer detection methods. With an accuracy of 0.31 nM, polyadenine can be quantified with fluorescent spherical nucleic acid labeled at the end of the mRNA in AuNPs [78]. This is therefore considered a precision medicine strategy to predict diseases by quantifying associated mRNAs. Among the major enzymes involved in DNA replication and repair is topoisomerase 1/2, which is implicated in cancer development or resistance to chemotherapy [79]. As a result of the labeling of a probe capable of detecting topoisomerase 1/2 in AuNPs, the absorbance ratio in a gold-aggregating assay can be calculated to determine the amount of AuNPs [79]. Likewise, quantitative measurements of biomarkers associated with cancer, such as glypican-1, facilitate the diagnosis of pancreatic cancer [80]. In Li’s work, glypican-1 was found to be pre-amplifiable when using the catalytic hairpin assembly and point-of-care-testing methods. Capturing glypican-1 mRNA through AuNPs is performed with paper-based strips with a 100 fM sensitivity [80].

In addition to regulating the transcription of DNA into RNA, a noncoding RNA can also regulate the translation of mRNA. The use of microRNA (miRNA) is one of the most common methods for regulating mRNA, so detecting the level of miRNA is considered a means of diagnosing diseases. It is possible to successfully capture miRNAs by designing corresponding DNA probe-AuNPs [81,82,83,84]. Various methods of detection have been used to quantify miRNAs in many studies. miRNA levels can be detected using electrochemical differential pulse voltammetry [85]. The results of Pothipor’s study suggested that the detection limit of miRNA-21 can reach as low as 0.020 fM after optimizing the electron transfer reaction between graphene and polypyrrole in AuNPs [85]. The introduction of MXene–MoS2 heterostructure nanocomposites was also found to improve electrochemistry and to form a catalytic hairpin assembly that can amplify the signal and enhance data detection [86]. Consequently, changing DNA structure may improve the efficiency of miRNA recognition. It has been shown that using three-dimensional or tetrahedral DNA nanostructure probes enhances electrochemical signals [87,88]. Furthermore, this DNA structure can also be applied to detect magnetic nanocomposites based on AuNPs [89]. Magnet-based devices for miRNA detection, such as the SERS-based biosensor, can be applied to serum and urine samples [90,91]. It is important to note that when using specific DNA structures and SERS, anticancer drugs can also be loaded to target miR21 for precision therapy [92].

### 4.3. Protein

As the final product of mRNA translation, a protein is a larger molecule than DNA and RNA, and conjugates with AuNPs can also be used to detect proteins. In cancer cells, telomerase, an enzyme responsible for protecting DNA replication from cell division, is overexpressed. It was found that the use of (TTAGGG)n sequences and AuNPs as probes could be used to diagnose cancer cells or the telomerase level in a liquid biopsy as a method of cancer diagnosis [93,94,95]. As a similar approach, single-strand DNA-rhodamine 6G-AuNPs were designed to be cleaved by Flap endonuclease 1, another molecule related to DNA structure; the potential Flap was calculated by exposing the rhodamine 6G after cleavage at the Flap endonuclease 1 level [96,97]. In addition to being used to screen for specific cell types, the aptamer can also be used to identify specific proteins. The use of designed aptamer-AuNPs has been reported to be specific for diagnosing prostate cancer through a prostate-specific antigen [98]. In breast cancer, the aptamer-AuNP method can also detect HER2 [99], and surface plasmon resonance technology can improve this method’s detection sensitivity [100].

## 5. Biosensing of Cancer Biomarkers

As most molecules can only be quantified through precision instruments, many studies have primarily focused on optical and enzyme-dependent biosensing to reduce detection time and cost. These two methods’ numerous applications in detecting cancer biomarkers are discussed in this section.

### 5.1. Optical Biosensing

Many studies have identified and determined the identity of generated AuNPs through their spectral differences since AuNPs require different media for ultraviolet–visible spectroscopy. Most studies have used electrochemical responses to assess whether targets are captured [101]. However, it should be noted that optical property can be used to facilitate detection and interpretation, thus reducing the requirements for equipment level or response time. Based on the surface plasmon resonance of AuNPs formed by incident photon frequency and free electrons, state changes can occur between dispersion and aggregation and are accompanied by color changes [102]. For this reason, colorimetric assays are used to interpret data. According to Miti’s study, localized surface plasmon resonance can be used to detect miR-17 levels in cancer cells [103]. It is also possible to combine it with the upconversion emission nanoparticle NaYF_4_ to amplify the signal for detecting miR-21-AuNPs [104]. This application can be used to detect not only miRNA sequences but also target miRNAs for the delivery of drugs. Introducing nanocomposite elements with AuNPs can not only enable the detection of miR-21 but also allow doxorubicin to be released through localized surface plasmon resonance to achieve therapeutic effects. miR-21 levels and cell death incidence can also be determined via color conversion. A nanocomposite that undergoes electron transfer can also be detected via electrochemiluminescence [105]. In Zhang’s study, Ru(bpy)32+ and boron nitride quantum dots were used to detect miR-21-AuNPs, and the resonance energy transfer was quantified through electrochemiluminescence [106,107].

Fluorescence labeling has also been employed as a further method of identifying AuNP probe signals. In Li’s research, miR-21 and miR-141 were simultaneously labeled with two fluorescent dyes on AuNPs to hybridize them with intracellular microRNA. The fluorescent signal changed when the target RNA was bound to the probe. Due to how flares are made, diagnostic sensitivity can be achieved [108]. Labeling DNA sequences with fluorescent signals can also be used to detect proteins. There is evidence that labeling Cy5 with (TTAGGG)n sequence-AuNPs can be combined with surface-enhanced Raman spectroscopy (SERS) to identify the telomerase level in tumor cells [109]. Moreover, SERS and AuNPs, in combination with doxorubicin in DNA sequence, were shown to detect miR-21 and drug release [92]. With fluorescence resonance energy transfer, specific proteins from a mixture can be detected using N-doped carbon dots if there is an interaction between the target protein and the carbon dots [110]. Using antibodies to recognize specific DNA sequences can also increase the signal. A prostate-specific antigen, kallikrein-3, is considered to be useful for the diagnosis of prostate cancer. By simultaneously labeling aptamer-AuNPs with streptavidin and anti-kallikrein-3-AuNPs, kallikrein-3 can be effectively detected. It is noteworthy that the detected signal can also be embedded in cellulose paper and read with a smartphone [111].

### 5.2. Enzyme-Dependent Biosensing

The enzyme-dependent biosensing method uses a catalytic reaction or formation of a structure when the target and probe are combined, resulting in the cleavage of a specific enzyme. The methylation of DNA is the most common detection method. Various restriction enzymes may cleave a methylated DNA sequence to reveal signal differences. It is also possible to apply a similar approach to the incidence of miRNA detection. Upon interaction between the target miRNA and the probe, a structure can be formed that can be recognized by a cleaved, duplex-specific nuclease. A specific antibody with a label can identify the cleaved structure and interpret the structure using methods such as electrochemistry [112]. It has been shown that this duplex-specific nuclease approach can be used in conjunction with fluorescence [113,114], surface plasmon resonance [115], or SERS [116] for the identification of signals in other ways.

The development of methods using optical and enzyme-dependent biosensing has also been gradual. According to Zhang’s research, noncoding RNAs can be bound and cleaved by exonuclease III, and the cleaved products can release signal DNA and be detected by fluorescence resonance energy transfer [117]. A prostate-specific antigen was used in Wang’s experiment to design a unique sequence that would be vulnerable to CRISPR-Cas12a cleavage. This method produces a product that can be quantified via the colorimetric method and used to diagnose prostate cancer [118].

## 6. Clinical Applications of Cancer Based AuNP Biomarkers

Based on the above discussion, AuNPs can distinguish between single bases and therefore be applied to detect DNA methylation. Accordingly, this section discusses single-point mutations and single-nucleotide polymorphisms as related diagnostic cases.

### 6.1. Single-Point Mutations or Single-Nucleotide Polymorphism (SNP)

AuNPs have also been used for the detection of single-nucleotide polymorphisms [119]. Analyses are primarily conducted by hybridizing a DNA microarray with single-strand DNA-AuNPs, and interpretation is obtained through analysis of various signals, including resonance light scattering [120] and SERS [121].

A single-point mutation is also a type of single-nucleotide polymorphism that can lead to the continuous activation of specific genes, ultimately resulting in a mutation. As a result of Park’s work, it was found that the mismatch recognition protein, the MutS protein, can recognize mutations of the KRAS gene. Combining MutS-AuNPs with sequence hybridization can change the resonance frequency detected by a microcantilever resonator [122]. It can also be applied to liquid biopsy-related incidents, including the detection of circulating tumor cells and the detection of ctDNA for the diagnosis of cancer [123,124,125].

### 6.2. Exon or Gene Copy-Number Changes Detection

There is an association between cancer progression and mutations in the EGFR and the development of drug resistance. Mainly, mutations in exon 19/21 are often detected in tissues and cells related to lung cancer. Via the hybridization of specific probe-AuNPs with a target sequence, AuNPs can be changed between dispersed and aggregated states that colorimetric assays can detect and applied to different tissues, including tumor cells and liquid biopsies [126,127].

### 6.3. Protein Structural Modifications Detection

Proteins such as transcription factors can be detected using AuNPs. This is accomplished by designing a suitable ligand or aptamer. It has been demonstrated that a protein–captured target interaction product can change signaling through fluorescence resonance energy transfer [110]. Additionally, a proper ligand can also be utilized in high-throughput screening to identify special drug candidates for alternative treatments [128]. Interestingly, a similar detection method was recently used to identify differences in proteins subjected to glycosylation [129].

## 7. Conclusions

The use of nanoparticles for bio-detection, drug delivery, and drug screening has become a new research direction for scientists. In addition, the combination of semiconductor etching technology and biomaterials can be used for biological disease detection or drug screening. Due to the unique spectroscopic properties of AuNPs, more detection methods can be established to perform a variety of analytical experiments on a single biochip. In pharmaceutical science and biotechnology, the rapid detection and reading of different DNA sequences is a more compelling development target. Combining DNA sequences with fluorescent probes, gold nanoparticles, and chemical luminescence has partially replaced old radio-isotope detection methods. The development of nanotechnology will bring unlimited opportunities for the development of life sciences, mainly regarding: (1) the study of the structure and function of various intracellular organelles (e.g., granulosa and nucleus) at the nanoscale in terms of the exchange of materials, energy, and information between cells and organisms; (2) biological response mechanisms in repair, replication, and regulation, as well as the development of molecular engineering (including the use of AuNP biomolecular robots), based on biological principles; (3) nanoscale imaging techniques, such as optical coherence chromatography (OCT), which is known by scientists as the “molecular radar” and has a resolution of 1 micron (which is thousands of times higher than the precision of available chromatography and nuclear magnetic resonance techniques), can conduct the dynamic imaging of living cells in living organisms 2000 times per second, and observe the dynamics of living cells without killing them as with X-ray, general chromatography, and MRI; (4) laser single-atom molecular detection, which is also ultra-sensitive and can accurately acquire one of the 100 billion atoms or molecules in gaseous materials at the single-atom molecular level; and (5) microprobe technology, which can be implanted into different parts of human body according to different research purposes and can also run with the blood in the body so that various types of biological information can be delivered to the external recording device at any time. Microprobe technology has the potential to become a standard tool for life science research in the 21st century. The development of nanotechnology will bring infinite hope to the development of biotechnology as it can be used to investigate the activities of biomolecules in the body to investigate issues affecting human health.

## Figures and Tables

**Figure 1 molecules-28-00364-f001:**
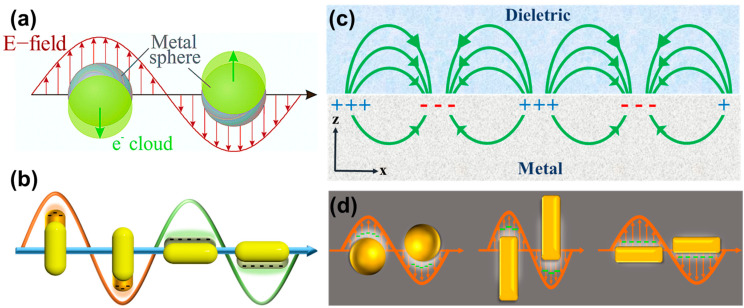
Schematic diagrams: (**a**) surface plasmon polariton. (**b**) Localized surface plasmon resonance. (**c**) A dipole centered on a metal sphere with a radius r smaller than the incident wavelength was used to oscillate metallic nanoparticles that were subjected to an electromagnetic field and oscillated by the dipole. (**d**) Excitation of AuNPs by incident light results in surface plasmon resonance. The surface plasmon resonance oscillating along different axes comprises the short-axis absorption peak and the long-axis absorption peak.

**Figure 2 molecules-28-00364-f002:**
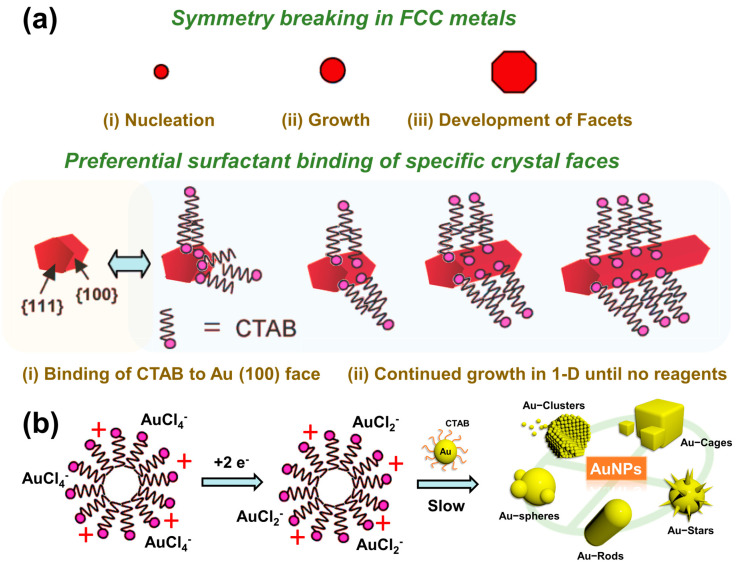
AuNP synthesis method. (**a**) In the surfactant-preferential-binding-directed growth mechanism, because CTAB protects the {100} side of the crystalline species, the subsequently added gold ions can only be stacked and grown on the unprotected {111} side, resulting in AuNPs. (**b**) Electric-field-directed growth mechanism, in which AuCl_4_-substituted Br is connected to the CTAB microcell and reacts with the CTAB-coated crystalline species.

**Figure 3 molecules-28-00364-f003:**
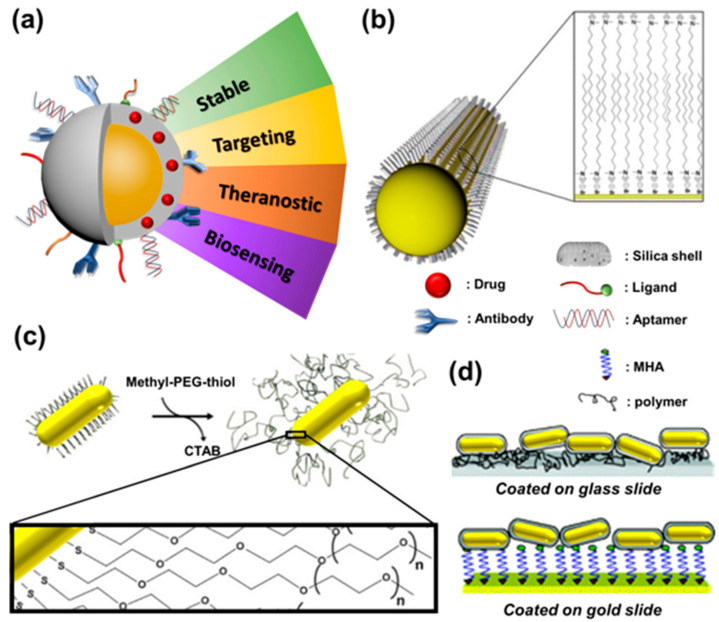
Surface modification of AuNPs. (**a**) Application of AuNPs in the biomedical field after surface modification. (**b**) Schematic diagram of the interface active agent CTAB adsorbed on the surface of AuNPs with a bilayer structure. (**c**) Schematic diagram of a modified polymer on AuNPs’ surface (n means the numbers of polymer molecules). (**d**) Layer-by-layer modification of charged polymer on AuNPs’ surface for selective adhesion to the substrate.

**Figure 4 molecules-28-00364-f004:**
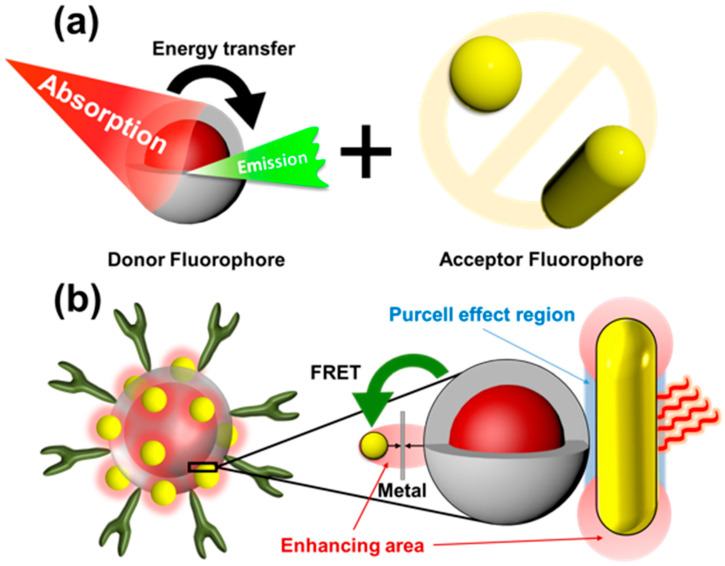
(**a**) The upconverted nanoparticles (donor fluorophore) are affected by the surface plasmon resonance (SPR) of AuNPs. (**b**) The FRET effect enhances the absorption of the excitation source, and the Purcell effect enhances the emission intensity.

**Figure 5 molecules-28-00364-f005:**
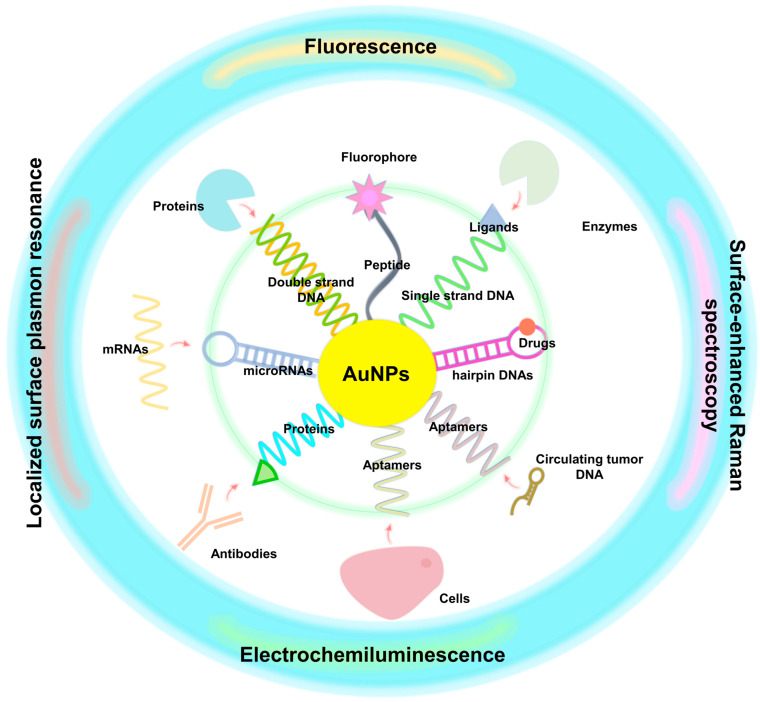
The development of cross-disciplinary nanotechnology has been vigorous, and one critical application is in biosensors produced by combining the semiconductor industry and biotechnology. AuNPs induce more biomolecular bonds, such as those of DNA, RNA, protein, and drugs to achieve the benefit of expanding the output signal. In addition, immobilization techniques can also be applied to a novel metal–semiconductor–metal material wafer. Using the chemiluminescence reaction in the system, the photons are converted into electronic signals to measure target biomolecules.

## Data Availability

Not applicable.

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
