# Peer review of "Gold Nanoparticles as a Biosensor for Cancer Biomarker Determination"

_molecules, 2023, doi:10.3390/molecules28010364_

Round 1

Reviewer 1 Report

The review paper described the application of AuNPs in fabrication of biosensors. I recommend a few considerations to improve the integrity of review contents and also present the novel research studies in this field study.   

The authors should describe the citrate mediated synthesis method in section 2.1. The brief description of this process should be indicated in this section.

In section 3.3.1, the FRET based inhibition mechanisms which results to fluorescence recovery processes should be indicated with reference to recently published manuscripts such as 10.1016/j.foodchem.2022.134212, 10.1016/j.saa.2022.121582.

Another section for colorimetric detection of bio analytes should be added to section 3.3 to provide information about the novel platforms for naked eye detection with citing related publication such as 10.1007/s00604-021-05111-6, 10.1016/j.bios.2022.114889 and 10.1088/1361-6528/ac23f7.

In Fig. 5 which molecules are attached to AuNPs surface as the linkers. Are they DNA or RNA strands? some of them are single strand and some other are single strand!

Line 364: The sentence “Such modification dysregulation leads to changes in specific genes” is not meaningful and should be changed to “such modification leads to dysregulation and abnormal expression of specific genes”.

Line 422: Similar nanocomposites such as KRAS can also be used as biomarkers for cancer. KRAS is the cancer gene not nanocomposite! It should be corrected.

Author Response

The review paper described the application of AuNPs in fabrication of biosensors. I recommend a few considerations to improve the integrity of review contents and also present the novel research studies in this field study.  
Ans: We thank the Referee for the time taken to review our work and for the constructive comments.

The authors should describe the citrate mediated synthesis method in section 2.1. The brief description of this process should be indicated in this section.

Ans: We thank the reviewer for their kind comments. The citrate-mediated synthesis method has been added before the CTAB process for your reference.

As a result, the following corrections have been made:

…In the beginning, Turjevich’s team used sodium citrate as a reducing agent and interfacial reactive agent to reduce HAuCl4 from Au3+ to Au by hydrothermal method. It synthesized many gold nanoparticles of different sizes under different reaction parameters. In the aqueous phase, the AuCl4- atoms are reduced by sodium citrate to form gold atoms, which then aggregate to form gold nanoparticles. The negatively charged citrate ions play the reducing agent and capping agent. The size of the particles is controlled by the ratio of gold ions to sodium citrate with the heating time of the reaction. At this time, the gold nanoparticles are protected by the negatively charged citrate on the outside of the particles and are stably stored in the aqueous solution.

Please refer to page 3, line 95-104,

2.1.  Surfactant-preferential-binding-directed growth section.

In section 3.3.1, the FRET based inhibition mechanisms which results to fluorescence recovery processes should be indicated with reference to recently published manuscripts such as 10.1016/j.foodchem.2022.134212, 10.1016/j.saa.2022.121582.

Ans: Thank you for sharing the critical references with us. Those FRET-related papers can help us to give more information on FRET-based inhibition mechanisms. Please kindly check the updated references in the manuscript as follows:

Ref.

  1. Dadmehr, M.; Shahi, S. C.; Malekkiani, M.; Korouzhdehi, B.; Tavassoli, A., A stem-loop like aptasensor for sensitive detection of aflatoxin based on graphene oxide/AuNPs nanocomposite platform. Food Chem 2023, 402.
  2. Hu, A. Q.; Chen, G. Q.; Yang, T. Q.; Ma, C. Q.; Li, L.; Gao, H.; Gu, J.; Zhu, C.; Wu, Y. M.; Li, X. L.; Wei, Y. T.; Huang, A. L.; Qiu, X. Q.; Xu, J. Z.; Shen, J. L.; Zhong, L. Y., A fluorescent probe based on FRET effect between carbon nanodots and gold nanoparticles for sensitive detection of thiourea. Spectrochim Acta A 2022, 281.

Please refer to page 20, line 776-795, References section.

Another section for colorimetric detection of bio analytes should be added to section 3.3 to provide information about the novel platforms for naked eye detection with citing related publication such as 10.1007/s00604-021-05111-6, 10.1016/j.bios.2022.114889 and 10.1088/1361-6528/ac23f7.

Ans: Thanks to the reviewer for sharing the critical references with us. We have added a section to discuss colorimetric detection of biological analytes. Those novel platforms can be used to support the concept of biosensing assays.

As a result, the following corrections have been made:

3.3.3 Fluorescent and colorimetric assay

AuNPs have unique physical and chemical properties, which make them easier to be novel chemical and biological sensors. In the past ten years of research, AuNPs have been widely used as sensors and in detecting heavy metals. Among them, the colorimetric method is a common and convenient detection method. The content of the component to be tested is determined through the solution’s color depth, and the sample’s concentration is analyzed by optical and photoelectric colorimetry. Both methods are calculated through the Beer–Lambert law (Equation 3.2). For example, DNA-functionalized gold nanoparticles can be used to detect lead ions with a colorimetric method: DNase-based functionalized gold nanoparticles; the sensor’s detection range can be from 3 nM to 1 μM [46]. However, DNA molecule synthesis and chemical modification are very complex and expensive.

…equation 3.2

(A is the absorbance;  is the molar attenuation coefficient or absorptivity of the attenuating species; l is the optical path length in cm; c is the concentration of the attenuating species)

However, colorimetric techniques tend to be less sensitive and less selective. To effectively improve its sensitivity and resolution, recent studies have used composite materials to enhance the accuracy of colorimetry. In current literature, AuNPs are used as a heavy metal sensor at different pH values. Mercury ions and lead ions will form mercury-gold and lead-gold alloys on the surface of AuNPs [47]. In the hydrogen peroxide (H2O2) presence, the fluorescent substrate is catalyzed with peroxidase-like properties. In order to make the sensor more stable to heat, pH, and salt, Li [48] et al. designed a fluorescent and colorimetric dual-modal sensor based on AuNPs prepared using carbon dots (CDs) to detect the concentration of Cu2+ and Hg2+ in water. Besides environmental heavy metal detection, AuNPs can also be used to analyze cancer biomarkers through dual-mode fluorescence colorimetry. Dadmehr [49] et al. used AuNPs grafted with gelatin and combined with gold nanoclusters to detect matrix metalloproteinase. Food samples can also be detected by AuNPs colorimetric method. Shahi [50] et al. used gelatin-functionalized AuNPs (AuNPs@gelatin) and applied a competitive colorimetric assay to detect aflatoxin B1 (AFB1) and measure the concentration of this carcinogen in food samples.

Please also kindly check the updated references in the manuscript as follows:

Ref.

  1. Li, Y. X.; Tang, L.; Zhu, C. X.; Liu, X. Y.; Wang, X.; Liu, Y., Fluorescent and colorimetric assay for determination of Cu(II) and Hg(II) using AuNPs reduced and wrapped by carbon dots. Microchim Acta 2022, 189, (1).
  2. Dadmehr, M.; Mortezaei, M.; Korouzhdehi, B., Dual mode fluorometric and colorimetric detection of matrix metalloproteinase MMP-9 as a cancer biomarker based on AuNPs@gelatin/ AuNCs nanocomposite. Biosens Bioelectron 2023, 220.
  3. Shahi, S. C.; Dadmehr, M.; Korouzhdehi, B.; Tavassoli, A., A novel colorimetric biosensor for sensitive detection of aflatoxin mediated by bacterial enzymatic reaction in saffron samples. Nanotechnology 2021, 32, (50).

Please refer to page 10-11, line 355-385, 3.3.3 Fluorescent and colorimetric assay section.

In Fig. 5 which molecules are attached to AuNPs surface as the linkers. Are they DNA or RNA strands? some of them are single strand and some other are single strand!

Ans: We thank this Referee for their constructive suggestion. We agree with the Referee’s comments. The Figure5 has been modified in accordance with the Referee’s recommendations.

As a result, the following corrections have been made:

Please refer to the page 12, line 398, Figure 5.

Line 364: The sentence “Such modification dysregulation leads to changes in specific genes” is not meaningful and should be changed to “such modification leads to dysregulation and abnormal expression of specific genes”.

Ans: We appreciate the Referee for raising the valuable questions. The revised descriptions have now been included.

As a result, the following corrections have been made:

Such modification leads to dysregulation and abnormal expression of specific genes.

Please refer to page 10, line 411, 4.1. DNA section.

Line 422: Similar nanocomposites such as KRAS can also be used as biomarkers for cancer. KRAS is the cancer gene not nanocomposite! It should be corrected.

Ans: We thank the Referee for bringing up this important point and agree with the Referee’s corrections. According to the Referee’s recommendation, we have revised the description.

As a result, the following corrections have been made:

Similar oncogenes such as KRAS can also be used as biomarkers for cancer diagnosis.

Please refer to page 11, line 470, 4.1. DNA section.

We thank the Referee for this important observation and related comments as well as the professional Referee’s work on our manuscript. We hope our revised manuscript is acceptable for publication in Molecules.

Reviewer 2 Report

In this review, the authors reviewed completely the applications of gold nanotechnology in cancer biomarker determination. On one hand, the working mechanisms of gold nanoparticles biosensors have been introduced detailly. On the other hand, the authors commented on clinical biomarker for early cancer detection, etc. Finally, the authors analyzed the clinical applications of cancer based on gold nanoparticles biosensors. The English writing of this review is very well. Additionally, the structure of this review is clear, and this review can promote researchers' rapid understanding of the latest progress in this field. Thus, I recommend that this review should be published. My comments are as follow.

(1) According to the contents, the Line 47 and 48 do not become a paragraph independently. This paragraph should be placed at the end of the paragraph of above it.

(2) In Line 185, the format of the equation may need to be changed.

Author Response

In this review, the authors reviewed completely the applications of gold nanotechnology in cancer biomarker determination. On one hand, the working mechanisms of gold nanoparticles biosensors have been introduced detailly. On the other hand, the authors commented on clinical biomarker for early cancer detection, etc. Finally, the authors analyzed the clinical applications of cancer based on gold nanoparticles biosensors. The English writing of this review is very well. Additionally, the structure of this review is clear, and this review can promote researchers’ rapid understanding of the latest progress in this field. Thus, I recommend that this review should be published. My comments are as follow.
Ans: We deeply appreciate this Referee for the positive and insightful suggestions.

(1) According to the contents, the Line 47 and 48 do not become a paragraph independently. This paragraph should be placed at the end of the paragraph of above it.

Ans: We thank the reviewer for the comments. Sorry to make a mistake in the separation of paragraphs. Lines 47 and 48 have been adjusted. Please kindly check the revised “Introduction” with a red mark.  

Please refer to page 2, line 47-48, Introduction section.

(2) In Line 185, the format of the equation may need to be changed.

Ans: Thank you for your kind suggestion. We have revised the equation format and rearranged the explanation to clarify the equation.

As a result, the following corrections have been made:

…equation 3.1

sp: surface plasmon resonance wavelength; n: free carrier density; m0: effective mass of charge carrier ; ε0: dielectic constant offree space; ; εm: medium dielectric constant; ε: high dielectric constant of medium; κ: geometry of nanomaterials ; and Γ: damping constant)

Please refer to page 5, lines 192 to 197, the section of “3.1 Localized surface plasmon resonance (LSPR).”

We thank the Referee for this important observation and related comments as well as the professional Referee’s work on our manuscript. We hope our revised manuscript is acceptable for publication in Molecules.

Reviewer 3 Report

The Review is devoted to a topical issue related to the description of the unique optical properties of gold nanomaterials, the synthesis and biological characteristics of AuNPs, and their applications as clinical biomarkers for early cancer detection. In this regard, the title of the review does not quite correspond to its content “Gold nanoparticles as a biosensor for cancer biomarker determination”, where only half of the second half of the text is related to “biosensor for cancer biomarker determination”, and the first half is devoted to the work on the principles of obtaining and operating AuNPs. Undoubtedly, applications of AuNPs as clinical biomarkers for early cancer detection are relevant, and sound advantageous for the title of the review; however, it is somewhat misleading to the reader, since it does not fully reflect the content. Reword the title of the review (expanding the topics covered) so that it also reflects the material described in the 2nd and 3rd sections. There are only 4 references in the Introduction for more than 50 lines. This is extremely little to confirm and illustrate the stated phrases. I think that at least 15-20 references are necessary for the Introduction of Review paper. Add them to the appropriate places in the text of the Introduction.

In general, the review is of particular interest to readers, the presented figures are original, and the review accumulates important information related to the properties and application of AuNPs. This review may be published in the journal Molecules, after correcting the comments made.

Comments

Lines 13-16

“The combination of plasmonic resonance, biochemistry, and optoelectronic engineering has opened up the detection of molecules and the possibility of atoms, medical research at the cellular level will have considerable application potential.”

Reframe the proposal. In the current format, it is difficult to reconcile. The first part is in the present tense, the second part is in the future tense.

Lines 35-37

“When the free electrons are influenced by electromagnetic radiation, an illustration of how the electric field looks in section, they are unequally distributed, i.e., an electric field with a deviating electric field causes negatively charged electrons to move in that direction.”

Reframe the proposal. Break this sentence into two simpler ones. In the current version, it is too difficult to understand.

Lines 37-38

In these lines you have used "that direction" twice. Specify which direction.

Line 50

Specify - what and / or where there are "shortcomings"

Figure 2.

Change the design of Figure 2 slightly by replacing the additional characters a, b, c (for Fig2a part 1) and a, b (or Fig2a part 2) with Arabic, Roman or some other numbering different from the internal numbering of Figure 2 (where a, b used for Fig1a, Fig1b). In the current design, there is some confusion when matching the notation.

Figure 5.

Reformulate the caption for Figure 5. In the current format, the caption for this figure is a discussion on a general topic and can serve as a caption for numerous similar figures. Specify in the caption to this figure the elements depicted in the figure itself.

Also expand the abbreviations used in the figure. For example, specify that ctDNA is Circulating tumor DNA.

In addition, the schematic nature of the structures originating from AuNPs is not clear. For DNA it is a double helix, for mRNA and miRNA it is a hairpin, for all other cases it is a helix. Shouldn't ctDNA also double helix? Maybe it's better to introduce a single type of arrows for all biological structures? Or for each structure to use an individual type of arrow? In the current version, the scheme looks unfinished, where the authors divide their outgoing AuNPs with arrows into DNA, RNA, and everything else.

Think about it, try to come to a uniformity in the designation of arrows (either all the same, or all have a specific design).

Author Response

The Review is devoted to a topical issue related to the description of the unique optical properties of gold nanomaterials, the synthesis and biological characteristics of AuNPs, and their applications as clinical biomarkers for early cancer detection. In this regard, the title of the review does not quite correspond to its content “Gold nanoparticles as a biosensor for cancer biomarker determination”, where only half of the second half of the text is related to “biosensor for cancer biomarker determination”, and the first half is devoted to the work on the principles of obtaining and operating AuNPs. Undoubtedly, applications of AuNPs as clinical biomarkers for early cancer detection are relevant, and sound advantageous for the title of the review; however, it is somewhat misleading to the reader, since it does not fully reflect the content. Reword the title of the review (expanding the topics covered) so that it also reflects the material described in the 2nd and 3rd sections. There are only 4 references in the Introduction for more than 50 lines. This is extremely little to confirm and illustrate the stated phrases. I think that at least 15-20 references are necessary for the Introduction of Review paper. Add them to the appropriate places in the text of the Introduction.

In general, the review is of particular interest to readers, the presented figures are original, and the review accumulates important information related to the properties and application of AuNPs. This review may be published in the journal Molecules, after correcting the comments made.

Ans: We thank the Referee for the time taken to review our work and for the constructive comments.

Comments

Lines 13-16

“The combination of plasmonic resonance, biochemistry, and optoelectronic engineering has opened up the detection of molecules and the possibility of atoms, medical research at the cellular level will have considerable application potential.”

Reframe the proposal. In the current format, it is difficult to reconcile. The first part is in the present tense, the second part is in the future tense.

Ans: We thank the reviewer for pointing out the mistake in the abstract. The paragraph has been reframed to make sure the tense is uniform.

As a result, the following corrections have been made:

…The combination of plasmonic resonance, biochemistry, and optoelectronic engineering has increased the detection of molecules and the possibility of atoms. Those advantages bring medical research to the cellular level for application potential.

Please refer to page 1, lines 13 to 16, the section of “Abstract.”

Lines 35-37

“When the free electrons are influenced by electromagnetic radiation, an illustration of how the electric field looks in section, they are unequally distributed, i.e., an electric field with a deviating electric field causes negatively charged electrons to move in that direction.”

Reframe the proposal. Break this sentence into two simpler ones. In the current version, it is too difficult to understand.

Ans: We would like to thank the reviewer for pointing out the sentences’ unclear. Based on your kind suggestion, we have separated this sentence into two simpler ones.

As a result, the following corrections have been made:

The uneven distribution of electrons illustrates the electric field at different cross-sections when free electrons are affected by electromagnetic radiation. For example, deviating from an electric field causes negatively charged electrons to move in the direction of the electric field. Transiently induced dipoles are generated, resulting in the separation of free electrons from the metal nucleus.

Please refer to page 1, lines 35 to 39, and the section of “Introduction.”

Lines 37-38

In these lines you have used “that direction” twice. Specify which direction.

Ans: Thank the reviewer for pointing out the typos in the manuscript. We have rewritten the sentence for your reference.

As a result, the following corrections have been made:

The uneven distribution of electrons illustrates the electric field at different cross-sections when free electrons are affected by electromagnetic radiation. For example, deviating from an electric field causes negatively charged electrons to move in the direction of the electric field. Transiently induced dipoles are generated, resulting in the separation of free electrons from the metal nucleus.

Please refer to page 1, lines 35 to 39, and the section of “Introduction.”

Line 50

Specify - what and / or where there are “shortcomings”

Ans: We thank the reviewer for the suggestion. The shortcomings have been added in the manuscript to support that nanocomposites can bring more advantages than single materials.

As a result, the following corrections have been made:

For instance, AuNPs need light energy to promote SPR, which upconversion nanoparticles can provide.

Please refer to page 1, line 50-51, and the section of “Introduction.”

Figure 2.

Change the design of Figure 2 slightly by replacing the additional characters a, b, c (for Fig2a part 1) and a, b (or Fig2a part 2) with Arabic, Roman or some other numbering different from the internal numbering of Figure 2 (where a, b used for Fig1a, Fig1b). In the current design, there is some confusion when matching the notation.

Ans: We thank the reviewer for pointing the comments out to us. The characters of Figure 2 have been modified to avoid confusion when matching the notation.

As a result, the following corrections have been made:

Please refer to page 4, line 145, Figure 2.

Figure 5.

Reformulate the caption for Figure 5. In the current format, the caption for this figure is a discussion on a general topic and can serve as a caption for numerous similar figures. Specify in the caption to this figure the elements depicted in the figure itself.

Also expand the abbreviations used in the figure. For example, specify that ctDNA is Circulating tumor DNA.

In addition, the schematic nature of the structures originating from AuNPs is not clear. For DNA it is a double helix, for mRNA and miRNA it is a hairpin, for all other cases it is a helix. Shouldn’t ctDNA also double helix? Maybe it’s better to introduce a single type of arrows for all biological structures? Or for each structure to use an individual type of arrow? In the current version, the scheme looks unfinished, where the authors divide their outgoing AuNPs with arrows into DNA, RNA, and everything else.

Think about it, try to come to a uniformity in the designation of arrows (either all the same, or all have a specific design).

Ans: We thank the Referee for pointing this out to us. The Figure 5 has been modified to address the Referee’s concern by emphasizing the schematic nature of the structures originating from AuNPs and their components of detection.

As a result, the following corrections have been made:

Please refer to page 12, line 398, Figure 5.

We thank the Referee for this important observation and related comments as well as the professional Referee’s work on our manuscript. We hope our revised manuscript is acceptable for publication in Molecules.

Round 2

Reviewer 1 Report

After consideration and according to amendments I recommend this review paper for publication.